# Validation of a Cantonese Version of the Amsterdam-Nijmegen Everyday Language Test (CANELT): A functional approach

**Winsy Wing Sze Wong** *

Department of Chinese & Bilingual Studies, Faculty of Humanities, Hong Kong Polytechnic University, Hong Kong SAR

* winsyws.wong@polyu.edu.hk

## Abstract

### Background

The current study aimed to validate the Cantonese version of the Amsterdam-Nijmegen Everyday Language Test (CANELT), a functional communication assessment tool for Cantonese speakers with aphasia. A quantitative scoring method was adopted to examine the pragmatics and informativeness of the production of people with aphasia (PWA).

### Method

CANELT was translated from its English version with cultural adaptations. The performance on the 20-item CANELT collected from 56 PWA and 100 neurologically healthy Cantonese-speaking controls aged 30 to 79 years was orthographically transcribed. Scoring was based on the completeness of the main concepts produced in the preamble and subsequent elaborations, defined as Opening (O) and New Information (NI). Measures examining the validity and reliability were conducted.

### Results

An age effect was found in neurologically healthy controls, and therefore *z* scores were used for subsequent comparisons between neurologically healthy controls and PWA. The test showed strong evidence for known-group validity in both O [$\chi^2$ (2) = 95.2, *p* < .001] and NI [$\chi^2$ (2) = 100.4, *p* < .001]. A moderate to strong correlation was found between CANELT and standardized aphasia assessment tools, suggesting satisfactory concurrent validity. Reliability measures were excellent in terms of internal consistency (Cronbach's α of .95 for both 'O' and 'NI'), test-retest reliability (ICC = .96; *p* < .001), intra-rater reliability (ICC = 1.00; *p* < .001), and inter-rater reliability for O (ICC = .99; *p* < .001) and NI (ICC = .99; *p* < .001). Sensitivity and specificity for O are 97% and 76.8%, respectively, while for NI, a sensitivity of 95% and specificity of 91.1% were obtained.

### Conclusions

Measures on validity and reliability yielded promising results, suggesting CANELT as a useful and reliable functional communication assessment for PWA. Its application in managing PWA and potential areas for development are discussed.

**Data Availability Statement:** The raw data involved in the validation process has been uploaded to an open-access repository (https://osf.io/p3ehr/files/osfstorage/65bf0c659b32ca077097f691).

**Funding:** The author(s) received no specific funding for this work.

**Competing interests:** The authors have declared that no competing interests exist.

## Introduction

Aphasia, a language impairment impacting one's comprehension and production in spoken or written modality, is present in about one-third of the post-stroke population [1]. People with aphasia (PWA) have difficulties in producing spoken discourse in various daily situations, impacting their participation in social/vocational contexts. Discourse processing and its production, which is concisely summarized in Dipper et al [2] as 'a unit of language larger than a single clause, forming a meaningful unit of language, and used for a specific purpose of function', is vital to functional communication. The ability to communicate in different real-life contexts is regarded as one of the most important treatment goals in PWA [3]. Hence there is a pressed need to develop ecologically valid assessment and treatment tools at discourse level.

Several assessment tools for measuring functional communication, based on discourse production in real-life contexts, have been developed for research or clinical purposes. Examples include the Communication Activities of Daily Living (CADL) [4] and its more recent versions [5, 6], the Amsterdam-Nijmegen Test for Everyday Language (ANELT) [7], and the Scenario Test [8]. Among these assessment tools, ANELT evaluates functional communication via monologic discourse production. In the original Dutch version of ANELT, 20 daily life scenarios tapping PWA's verbal responses are included, with standardized procedures, normative data, and two parallel versions, each consisting of 10 scenarios. Sample scenarios include: 'You are now at the dry cleaners. You have come to pick this up and you get it back like this [present shirt with scorch mark]. What do you say?', 'The kids on the street are playing football in your yard. You have asked them before not to do that. You go outside and speak to the boys. What do you say?'. These scenarios resemble realistic everyday communication situations in which monologic discourse is elicited from the PWA to fulfill different naturalistic communication goals in a conversation, such as to request, inform, instruct, or persuade the listener (in this case, the experimenter). Therefore, ANELT is also regarded as a test for verbal functional communication [9], i.e., the ability to convey ideas independently and effectively [10] in everyday communication contexts.

Scoring of traditional ANELT is qualitative, based on two scales: Comprehensibility A-Scale and Intelligibility B-Scale. A-Scale refers to the content of the message conveyed by the PWA irrespective of its linguistic form, whereas B-Scale examines the perception of the utterance in such a way that words are recognizable regardless of their meaning [7]. A five-point rating ranging from very good to very bad is used by the administrators based on their subjective judgment. With satisfactory internal consistency, test-retest reliability, and inter-rater reliability of A-Scale as established by the developers of the test, they suggested ANELT be used as a reliable and valid assessment of the verbal adequacy of the PWA [7]. The A-Scale has been used predominantly in several aphasia treatment studies as a communication measure [9, 11] and an outcome measure of verbal adequacy in PWA speaking different languages such as Dutch [12], English [13] and Swedish [14].

For speakers of Chinese or Cantonese, standardized assessments on functional communication available for PWA in clinical or research contexts are rather limited. The Cantonese translated version of the American Speech-Language-Hearing Association Functional Assessment of Communication Skills (ASHA-FACS) [15] has been developed and validated [16] to evaluate PWA's communication independence and qualitative dimensions of communication based on clinician's observations in various naturalistic settings via a scale from 1 to 7. Despite its comprehensive coverage of social communication contexts, extra observation beyond the clinical setting is needed to evaluate the PWA's competence in communication, e.g., to exchange information via phone or to involve in a group discussion. Such detailed assessment is challenging in the local hospital setting, where only a 30-minute session for aphasia

evaluation is allowed [17]. Another available standardized assessment, the Main Concept Analysis (MCA) for oral discourse production [18], measures the informativeness and efficiency of verbal discourse production based on sequenced pictures. Nevertheless, the context (i.e., sequenced pictures) may not be naturalistic/typical in everyday communication. As Doedens and Meteyard [10; p. 939] suggested, the 'move from an in-vacuo task to a situated task' is necessary for functional assessment of PWA.

With regard to the lack of functional communication assessment to accommodate the above-mentioned limitations in rehabilitation settings in Hong Kong, an initiative to translate and develop a culturally appropriate ANELT in its Cantonese version was done by Law and Lo [19]. They took into consideration the quantitative scoring system proposed in Ruiter et al [20], as it is argued that a quantitative measure is more sensitive in detecting language performance over time among PWA [20, 21]. In Law and Lo [19], speakers' performance was evaluated based on the number of content units (CU) produced. CU is defined as "a group of information that was always expressed as a unit by normal speakers" [22 p 30]. Instead of adopting the rating-based evaluation of the original version, Law and Lo [19] proposed a word-based quantitative approach in which an information word list was compiled for each scenario by including words spoken by at least 25% of 59 healthy Cantonese aged 60 years or above. The intended outcome was to establish a quantitative yet transcription-less tool for assessing functional communication of Cantonese PWA by allowing clinicians to look up the informative word from the list during scoring PWA's responses online. However, a closer examination of the word lists generated from such a criterion revealed that it consisted of many function words, including 'then', 'is', which are non-specific to the scenarios and contribute little to the informativeness of the messages conveyed. Moreover, the open-ended nature of the scenarios presented in CANELT has resulted in high variability in plausible responses, which may not be included in the pre-compiled and restricted word list suggested. For example, in one scenario, speakers were asked to give verbal expressions on how they would order a flower/fruit basket; acceptable responses might include a wide range of information indicating the selection of different flowers/fruits/decorations or specifications on details like budget, recipient's name/address. In sum, a quantitative scoring method that could capture the presence and completeness of the inherently diverse information while keeping the tool as a clinically viable and valid assessment for functional communication of Cantonese-speaking PWA is strongly needed.

Inspired by the quantitative ANELT scoring scale developed by Ruiter and her colleagues [20], Yip [23] proposed a quantitative version of CANELT based on the two pragmatic components routinely expressed by neurologically healthy controls: 1) 'Opening' (which is referred to as 'preamble' in Ruiter et al [20]), i.e., to initiate communication/a conversation; and 2) New Information (which is referred to as 'requests' in Ruiter et al [20]), i.e., to elaborate the communicative goal and to seek further actions/verbal responses from the communication partner. Informativeness was based on the relevant content produced to fulfill the respective components and was not restricted to a designated word list. Furthermore, to better evaluate the completeness of the information, which is usually impaired among many of the PWA, the idea of main concept analysis and its scoring criteria (which will be explained in detail in the Method section) was introduced. By comparing the performance of 89 neurological healthy controls and 26 PWA, Yip [23] found that most PWA did significantly poorer than neurologically healthy controls in both Opening and New Information. Following Yip's [23] work, Lor [24] aimed to validate CANELT and its two parallel versions based on 57 PWA and 57 age-matched neurologically healthy controls. While satisfactory results were yielded on various psychometric properties, Lor [24] pointed out that the age effect among neurologically healthy controls in both components might misclassify PWA/ neurologically healthy controls. However,

insufficient neurologically healthy controls across all age groups made using age-normed scores impossible.

### Research aim

The current study aimed to validate CANELT as a clinical tool for Cantonese-speaking PWA. Despite its similarity with ANELT in that both measure functional communication using monologic discourse elicited via daily contexts, CANELT differs from ANELT in terms of 1) its quantitative nature in judging the adequacy of verbal production; 2) its consideration of different pragmatic functions of PWA in the scoring criteria, 3) adaptations of some of the scenarios to ensure cultural appropriateness for Cantonese speaking PWA in Hong Kong. Thus, a thorough examination into the psychometric properties of CANELT concerning validity (including face, known-group, and concurrent validity, specificity, and sensitivity), reliability (including internal consistency, test-retest reliability, and inter/intra-rater reliability) was conducted, based on data collected from current and previous studies [19, 23, 24].

## Materials and methods

### Participants

A total of 56 PWA reported in Lor [24] was included in the current study in the validation process. The inclusion criteria of PWA were: i) premorbid fluent speakers of Cantonese, ii) a diagnosis of aphasia using the Cantonese version of Western Aphasia Battery (CAB) [25], with a cut-off aphasia quotient (AQ) of 96.4 or below, and iii) post-onset period of at least six months. Exclusion criteria included: i) co-existing moderate-to-severe motor speech disorders and ii) comorbidity of other neurological disorders such as Alzheimer's disease and Huntington's disease, based on the medical reports provided by the PWA or reports from their caregivers. Administration of assessment tasks was conducted either by the author, who is a qualified speech-language therapist, or by trained speech-language therapy students under the author's supervision. The trained students had prior clinical experience with the neurological population. The mean years of education received by the PWA was 10.1 years (s.d. = 3.5). They spoke Cantonese, a Chinese dialect as their first language, while seven of them spoke and read English in their workplace on a regular basis. One hundred neurologically healthy controls aged 30 to 79 years were included to establish the normative data of CANELT. Five age stratifications, i.e., 30–39, 40–49, 50–59, 60–69, and 70–79, were defined, with each group containing 20 neurologically healthy controls. While 29 of them were recruited in the current study, the rest were based on data collected by Law and Lo [n = 35; 19], Yip [n = 30; 23], and Lor [n = 6; 24]. Ethical approval was obtained from the Human Research Ethics Committee of the University of Hong Kong (EA210334). Subject recruitment period spanned from September 16, 2021 to August 20, 2022. Written informed consent was obtained from all participants. Summary statistics on the demographic information of the PWA and neurologically healthy controls are provided in Table 1.

### Materials and procedures

**Translation of CANELT.** A research agreement has been made with the publisher of ANELT for the translation of the English version of ANELT to Cantonese and the dissemination of research findings [26]. ANELT was initially translated into Cantonese by Law and Lo [19]. Subsequently, a backward translation into English was carried out by one of the authors of Law and Lo [19] to ensure consistency and equivalence [27]. A pilot test was administered on three healthy Cantonese-speaking adults aged between 48 and 59 years. Among the 20

**Table 1. Summary statistics on the demographic information of PWA and neurologically healthy controls.**

| | | Neurologically healthy controls(n = 100) | PWA (n = 56) |
|---|---|---|---|
| | | Age | |
| | | Mean (s.d.) | |
| Age group (Years) | 30–39 | 34.8 (2.61) | 36.7 (1.50) |
| | 40–49 | 45.3 (2.57) | 44.4 (2.72) |
| | 50–59 | 54.5 (2.61) | 54.8 (2.95) |
| | 60–69 | 63.8 (2.50) | 64.1 (3.12) |
| | 70–79 | 73.8 (3.48) | 73.0 (3.22) |
| Gender | Male | 42 | 38 |
| | Female | 58 | 18 |
| Aphasia subtypes of PWA | | | |
| Fluency | Fluent (n = 27) | | Non-fluent (n = 29) |
| Aphasia subtypes | Anomia (n = 24) | | Broca's (n = 13) |
| | Wernicke's (n = 3) | | Transcortical Motor (n = 11) |
| | | | Global (n = 5) |

NA = not applicable.

items, five of them were considered culturally inappropriate and were therefore modified to meet the local culture (see S1 Appendix for items being modified and the reasons).

**Administration of CANELT.** CANELT consisted of two practice trials and 20 test trials. Each trial was a verbal description of a daily life scenario, followed by a verbal prompt of "What do you say?". Whenever appropriate, real objects such as a stained t-shirt and a banknote, were presented (details given in S2 Appendix) to illustrate the context. Respondents were asked to give verbal responses by taking the role in the described scenario. No time limit was given; usually, one repetition of the scenario description and one general verbal prompt like "What else would you say?" would be provided by the administrator, who was either a trained speech-language therapy student or a speech-language therapist with experience in assessing PWA. All elicited responses were audiotaped for subsequent scoring. The same administration protocol was applied across the previously collected data [19, 23, 24] and in the current study.

**Scoring of CANELT.** The orthographic transcriptions compiled by Law and Lo [19], Yip [23], Lor [24], and the author of this report were used for scoring. Scoring was based on the system developed by Yip [23], in which two components including 'Opening' (O) and 'New Information' (NI), were adopted. The former referred to utterance(s) that contained information provided in the descriptions of the scenario, which served as a topic initiation, i.e., preambles, in a conversation. The latter referred to utterance(s) that gave additional information/ elaboration, which was not given explicitly in the scenario descriptions. Unlike Ruiter et al [20], in which the term 'requests' was used, we defined such additional information/elaboration as 'New Information'. It is because a variety of communicative functions might be expressed by the speaker, such as information seeking, rejection, and suggestion of follow-up requests. S2 Appendix illustrates some examples of 'Opening' and 'New Information'.

Verbal production considered under 'O' or 'NI' was scored with reference to the presence and degree of completeness of the information or main concept(s) expressed. According to Nicholas and Brookshire [28 p 148], a main concept was defined as a statement that provides "an outline of the gist or essential information portrayed in the stimulus . . . and should contain one and only one main verb". In other words, an utterance with the inclusion of both subject/object and predicate would be considered a complete O/NI. One point was given if the

information was considered present and complete, while 0.5 point was given to information that was present but incomplete. No credit was granted for the absence of O/NI. A maximum score of 40 was given to each participant (i.e., one score credited to O and NI of each scenario, respectively, for a total of 20 scenarios). As data collected and reported in Law and Lo [19] was previously analyzed on different criteria, Yip [23] re-analyzed the orthographic transcriptions based on her framework proposed. In other words, the same scoring method was applied across previous studies [23, 24] and the current study. Details of scoring criteria and examples are illustrated in S2 Appendix.

**Data analysis.** Different statistical measures were utilized in terms of 1) establishing normative data of neurologically healthy adult speakers of Cantonese on CANELT, 2) validating CANELT as an assessment tool for PWA, and 3) conducting measures of reliability of CANELT.

The raw total scores of 'O' and 'NI' obtained from the neurologically healthy controls were used to establish normative data and as well to investigate the possible effects of age on the performance of CANELT via ANOVA tests. If assumptions of normal distribution and homogeneity of variances were violated, Kruskal-Wallis one-way ANOVA tests would be conducted on scores of 'O' and 'NI' followed by Games-Howell posthoc tests for pairwise comparisons among different age groups. Performance of neurologically healthy controls and PWA on 'O' and 'NI' were then converted into z-scores, based on the normative data of the respective age group, for various validation procedures described below.

**Known-group validity.** Known-group validity is a form of construct validity used to test if a tool is able to discriminate distinct groups (in this case PWA vs. neurologically healthy controls). The group differences in 'O' and 'NI' scores of CANELT among neurologically healthy controls, fluent and non-fluent PWA (based on their CAB subtypes) were investigated via Kruskal-Willis one-way ANOVA and Dwass-Stell-Critchlow-Fligner (DSCF) pairwise comparisons.

**Concurrent validity.** Concurrent validity examines whether a new instrument can evaluate what it intends to assess by corelating the scores obtained with another established measure taken at the same time. The Pearson product-moment correlation coefficients between z-scores of 'O' and 'NI' of the PWA and two aphasia assessments available for Cantonese PWA, namely CAB (in terms of AQ; n = 56) and MCA (in terms of total MC scores; n = 48), were calculated to examine concurrent validity.

**Specificity and sensitivity.** Specificity and sensitivity are commonly used to assess test performance against the gold standard [29]. The former tests how well a tool can correctly identify every subject who does not have the target disorder (i.e., true negative) while the latter examines if the tool can identify every subject who has the target disorder (i.e., true positive). A receiver-operating characteristic (ROC) curve, a graphical representation of true positive rate (i.e., sensitivity) on the y-axis against false positive (1-specificity) on the x-axis, is usually used to determine the best cut-off for the test. The ability of CANELT to distinguish between PWA and neurologically healthy controls was evaluated. A true positive diagnosis was determined by CAB, with a total AQ of 96.4 below. Receiver operating characteristic (ROC) curve and Area Under the ROC Curve (AUC) were used based on the sample of 100 neurologically healthy controls and 56 PWA.

**Face validity.** Face validity involves an informal review of the test questions by non-experts, who evaluate their clarify, comprehensibility, and appropriateness for the target group [30]. A total of 20 participants, including nine neurologically healthy controls and 11 PWA who obtained a sub-AQ score of 7 or above in CAB, were interviewed. The criterion set for PWA was to ensure adequate comprehension ability for investigating face validity. After the administration of CANELT, they were interviewed to give ratings, based on a 3-point Likert

scale [31], to each of the questions regarding (i) its ability to assess one's communication ability and (ii) the naturalness of the situations in everyday communication and whether the situation had been encountered before in real life. Items with low face validity, i.e., 30% or more respondents rated the item as 'poor' in assessing one's communication ability, should be excluded [32].

**Test-retest/ inter- and intra-rater reliability.** Test-retest reliability aims to measure the consistency of results obtained from the same person on two or more assessments. To examine test-retest reliability, CANELT was administered twice to 18 participants (nine neurologically healthy controls and nine PWA) within a 7–14 days interval. The correlation between the two sub-scores in the two administrations of CANELT was examined via intraclass correlation coefficients (ICCs). Intra-rater reliability refers to the consistency of measurement conducted by the same rater using the same criterion at different times whereas inter-rater reliability examines the consistency of measurements taken by two or more raters on the same subject over a single. Intra-rater reliability was examined via calculating the ICCs of the nine PWA on 'O' and 'NI', as scored by the same experimenter Lor. On the other hand, inter-rater reliability on 'O' and 'NI' was evaluated based on the scores of nine PWA from Lor [24] and 10 participants (five neurologically healthy controls and five PWA) from Yip [23]. ICCs of the scores given by the two experimenters (i.e., Lor and Yip) were compared against those given by the author of the current study.

**Internal consistency.** Internal consistency measures the degree of homogeneity among the items on a test, i.e., whether they are intercorrelated and measure the same construct. To evaluate the internal consistency of test items in CANELT, Cronbach's alpha values of the 'Opening' and 'New Information' were calculated based on the 20 items collected from neurologically healthy controls and PWA.

## Results

Table 2 summarizes the performance (in terms of raw scores) of neurologically healthy controls and PWA in terms of mean, standard deviation, and range on the two CANELT measures in their respective age group. Since the assumption of normal distribution was violated in scores obtained from neurologically healthy controls while homogeneity of variance was not observed in scores obtained from PWA, Kruskal-Wallis one-way ANOVA was conducted, in which a significant age effect ($p < .001$) was observed (see results on post-hoc pairwise

**Table 2. Descriptive summary of the performance of the PWA and neurologically healthy controls on CANELT measures in raw scores.**

| Group | Age group | Opening (out of 20) | | | New Information (out of 20) | | |
|---|---|---|---|---|---|---|---|
| | | Mean | SD | Range | Mean | SD | Range |
| Controls | 30–39 | 19.4 | 0.93 | 16–20 | 19.1 | 1.05 | 16–20 |
| | 40–49 | 19.2 | 1.51 | 14–20 | 18.8 | 1.12 | 16–20 |
| | 50–59 | 19.0 | 1.26 | 16–20 | 18.0 | 1.75 | 13.5–20 |
| | 60–69 | 17.4 | 2.11 | 13–20 | 16.4 | 2.01 | 12–19 |
| | 70–79 | 17.1 | 2.13 | 13.5–20 | 15.7 | 2.56 | 12–20 |
| PWA | 30–39 | 9.71 | 5.71 | 0.5–18.0 | 8.43 | 5.09 | 0–15.5 |
| | 40–49 | 9.56 | 7.40 | 0–19.0 | 7.69 | 7.88 | 0–18.0 |
| | 50–59 | 9.20 | 6.29 | 0–18.0 | 7.09 | 5.97 | 0–16.0 |
| | 60–69 | 8.38 | 6.37 | 0–19.0 | 6.00 | 6.73 | 0–19.0 |
| | 70–79 | 7.83 | 6.01 | 0–16.0 | 5.83 | 5.52 | 0–11.5 |

PWA = People with aphasia

comparisons in S3 Appendix). Neurologically healthy controls aged between 30 and 59 years performed similarly on 'Opening' while their scores were significantly higher than those between 60 and 79 years. The trend observed in 'New Information' was somewhat similar. Those aged between 30 and 59 years obtained similar scores, while the younger groups (30–39 and 40–49 years) performed better than the older groups (60–69, 70–79). Controls aged between 50 and 59 years scored significantly higher than the oldest age group (70–79 years). Performance of the two oldest age groups (i.e., 60–69 and 70–79 years) did not differ in 'O' or 'NI'. Subsequently, the raw scores of all neurologically healthy controls and PWA were converted to z-scores where appropriate to examine various validity measures, including known-group validity, concurrent validity, specificity, and sensitivity.

## Face validity

Face validity was satisfactory in general. The respondents agreed that the scenarios of CANELT could assess one's communication ability. Among the 20 scenarios, scenario six (i.e., seeing a $100 banknote nearby a butcher's) had the poorest rating, with 25% of the respondents rated the item as 'unable to assess communication ability.' Since none of the items had more than 30% of the participants rated as 'unable to assess communication ability', all scenarios were retained.

When asked if the situations were encountered in their real life, in five out of 20 scenarios (i.e., 25%), more than 50% of the respondents indicated that they had not encountered the situations before. They included scenario 2 (ask the kids not to play football at the doorway), 4 (repairing the toe cap of the shoes), 8 (calling the management office about the neighbor's dog non-stop barking), 16 (talking to a friend about the arm cast), and 18 (calling the missing cat owner).

## Known-group validity

Kruskal-Willis one-way ANOVA revealed that the performance of neurologically healthy controls, fluent and non-fluent PWA was significantly different on both CANELT measures of Opening [$\chi^2$ (2) = 95.2, $p < .001$] and New Information [$\chi^2$ (2) = 100.4, $p < .001$]. DSCF pairwise comparisons showed that the control group scored significantly higher than both fluent and non-fluent PWA ($p < .001$) in both 'O' and 'NI' while fluent PWA performed better in both CANELT measures than non-fluent PWA ($p < .001$).

## Concurrent validity

Satisfactory results were obtained on measures of concurrent validity. A strong correlation was found between AQ in CAB in both 'O' ($r = .78$; $p < .001$) and 'NI' ($r = .76$; $p < .001$). A moderate correlation existed between z-scores of MCA and both 'O' ($r = .64$; $p < .001$) and 'NI' ($r = .62$; $p < .001$).

## Sensitivity and specificity

The ROC curves and AUC for 'O' and 'NI' are given in Fig 1. A cutoff score of -2.09 (in terms of z score) was set for 'Opening', which yielded a sensitivity and specificity of 97% and 76.79%, respectively. The Youden's index was .74 while the AUC was .95. As for 'New Information', the cutoff score was selected at a z-score of 1.61, with a sensitivity and specificity of 95% and 91.07%, respectively. A Youden's index of .84 was obtained while the AUC was .97. Since the cutoff scores were different across the age groups, the respective cutoff scores for the five age groups are given in Table 3.

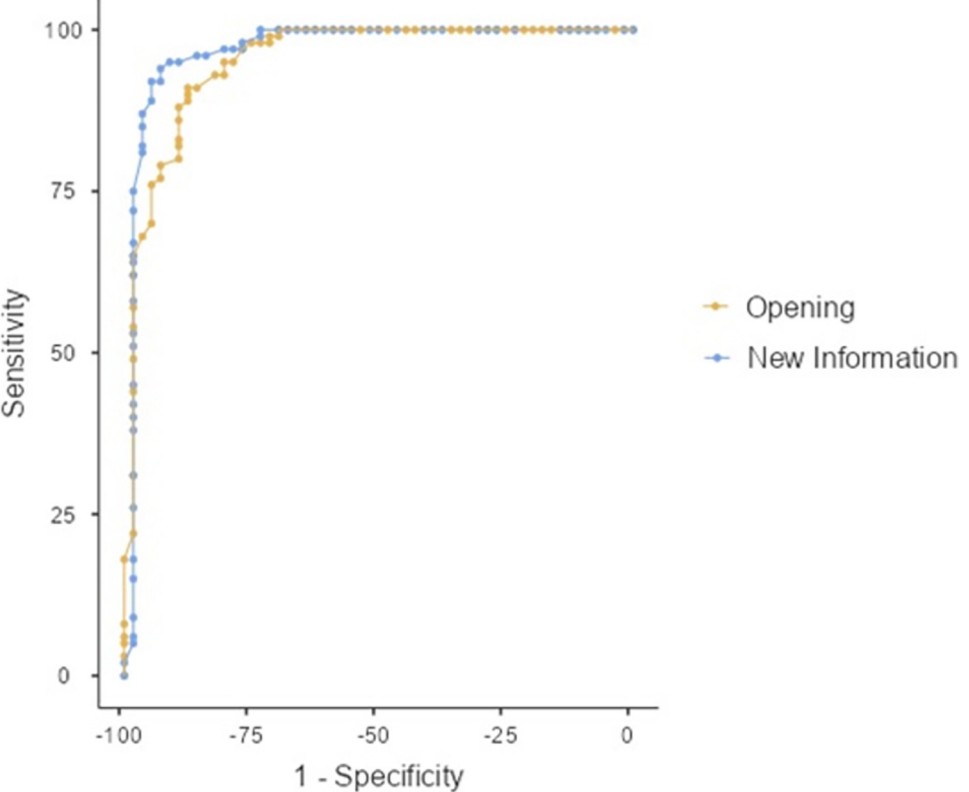

**Fig 1. ROC curves of 'Opening' and 'New Information' of CANELT.**

## Reliability measures

ICCs of test-retest reliability for 'Opening' and 'New Information' were .96 ($p < .001$) and .98 ($p < .001$), respectively. Intra-rater reliability was perfect for both 'O' and 'NI' (ICC = 1.00; $p < .001$), while the ICCs of inter-rater reliability for 'O' and 'NI' were .99 ($p < .001$) and 0.99 ($p < .001$), respectively. Regarding internal consistency, Cronbach's α of .95 was obtained for both 'O' and 'NI'. When the item with the lowest item-total correlation from 'Opening' or 'New Information' were excluded, Cronbach's alpha values were unaffected in either of the measures.

## Discussion

The current study aimed to validate a Cantonese version of the Amsterdam-Nijmegen Everyday Language Test, which serves as a diagnostic tool for functional communication of Cantonese-speaking PWA. Twenty culturally appropriate scenarios of daily communication have been adapted from its original Dutch version. One hundred neurologically healthy controls and 56 fluent and non-fluent PWA participated in the validation process. A scoring method

**Table 3. Cutoff Scores for 'Opening' and 'New Information' of CANELT across different age groups.**

|  | 30–39 years | 40–49 years | 50–59 years | 60–69 years | 70–79 years |
|---|---|---|---|---|---|
| **Opening** | **17.0** | **16.0** | **16.0** | **12.5** | **12.5** |
| New information | 17.0 | 16.5 | 15.0 | 13.0 | 11.5 |

that intends to capture the functional communication ability of PWA via a pragmatic and content-based approach has been adopted.

A series of validity and reliability measures have been conducted, which yielded satisfactory results. Face validity was satisfactory; all twenty items were considered adequate to assess daily communication. Scenarios frequently and less frequently encountered in daily life were also included. Known-group validity values for differentiating neurologically healthy controls from fluent and non-fluent PWA were high for both 'Opening' and 'New Information', indicating that both measures can discriminate against the three groups. Concurrent validity was also satisfactory, as the two measures of CANELT were positively correlated with two existing assessment tools of aphasia for Cantonese speakers, namely CAB and MCA. CANELT was also proved to be a reliable test for PWA, as supported by high test-retest, intra/inter-rater reliability, and internal consistency. Given the presence of an age effect in neurologically healthy controls, the use of $z$ scores as cutoff scores were adopted, which improved the sensitivity and specificity of CANELT. Based on the above findings, together with its relative ease in both administration and scoring, it is suggested that CANELT can be used as a valid and reliable diagnostic tool for assessing functional communication of Cantonese-speaking PWA.

Despite the difference in scoring method, CANELT, like the original qualitative [7] and quantitative [20] means of measurement, serves as a valid tool for assessing verbal communication of PWA. The current test did equally well in known-group validity measures when compared to both transcription-based and transcription-less qualitative scoring methods reported in Ruiter et al [33] ($p < .001$). In addition, the finding that non-fluent PWA scored significantly lower than the fluent PWA was somewhat similar to [15], in which fluent PWA demonstrated better functional communication than non-fluent PWA, as measured by the ASHA FACS.

Clinically speaking, the application of CANELT is suggested to be useful in managing Cantonese-speaking PWA. It can be conducted in a relatively short time and is easy to use in various clinical settings to reflect one's performance in various daily communication contexts. The scores in 'O' and 'NI' may help clinicians have a better understanding of the strengths and weaknesses of the client in different pragmatic functions, which may help in setting up appropriate goals for rehabilitation. Indeed, the current quantitative measure of communication functions, taken from a pragmatic point of view, revealed that the linguistic deficits of PWA would affect their pragmatic competence in conversation. This area has been less extensively evaluated/assessed in diagnostic tools for PWA. Although Ruiter et al [20] examined preambles and requests in their scoring criteria, PWA's performance on these two aspects was not evaluated separately. It is suggested that the current scoring system may better reflect the pragmatic competence of PWA.

## Limitations and suggestions for future research

CANELT, like its original Dutch version, primarily measures the effectiveness of verbal communication. Verbal efficiency was not considered. Although presenting real objects in some scenarios may facilitate comprehension, understanding the contexts is mainly through verbal means. Pictures or written words are not given to supplement verbal instructions. Acceptable output from the PWA, on the other hand, is essentially verbal. Hence PWA with severe receptive and expressive impairment might find the test too difficult to conduct. Their non-verbal responses such as gestures would not be captured and recognized under such scoring criteria. Further development of the test, similar to the Scenario Test [8] that examines multi-modal communication means such as written words, gestures, pictures, and communication aids, may help researchers and clinicians evaluate functional communication of PWA.

Another area of development of CANELT concerns its scoring efficiency and clinical application. A transcription-less quantitative analysis of ANELT has been developed by Ruiter et al [33], which yielded equally well psychometric properties as the transcription-based scoring. Since the scoring of participants' responses in the present study was based on orthographic transcription, one may wonder if a transcription-free scoring might allow clinicians working in hectic settings (such as an acute hospital) to evaluate patients' communication efficiently. Given its similarity to the scoring criteria used in Ruiter et al [33], it is believed that clinicians, after some basic training and practice, could be capable of evaluating PWA's performance efficiently without orthographic transcription. Besides, its sensitivity in monitoring treatment-induced changes is certainly worth investigating. Even though the Scenario Test has already been selected as the preferred instrument for assessing communication in the core outcome set for the Research Outcome Measurement in Aphasia (ROMA-2) [34], the results from CANELT may help in the development and validation of the Scenario Test for Cantonese-speaking population.

An aging effect on the performance of CANELT has been observed, in which the younger adults (below 60 years) performed better in both 'Opening' and 'New Information' than the older adults (60 years or above). The trend observed seems to be consistent with some of the previous reports on the effects of aging on discourse production (see Kintz et al [35] for a review and **discussion**). For example, Marini et al. [36] found that global coherence in picture description started to show a decline in neurologically healthy participants aged around 60 years old. A decline in cognition (especially in executive functions, memory, and attention) might account for the difference (see Olea Santos [37] for a summary and discussion). However, such an age difference identified in the current study should be treated with caution. Education level was not controlled across the age groups. Furthermore, detailed neuropsychological profiles of the participants were unavailable, making further investigation into the relation between cognition and performance on CANELT impossible. Similarly, the inferior performance of PWA to neurologically healthy controls on both CANELT measures might be not only attributed to their linguistic deficits alone. Non-linguistic cognitive impairments have been commonly found in PWA while their impacts on language performance have been explored and discussed in previous reports [38, 39]. In order to produce an appropriate response in 'O', one has to select the relevant information and keep it in verbal short-term memory, whereas for 'NI', cognitive skills such as working memory, planning, and problem-solving will be involved. Previous studies [9] have highlighted the correlation between brain lesions and performance on the Scenario Test; the relation between non-linguistic processing and performance on the two CANELT measures in neurologically healthy adults across the lifespan and among PWA deserves further investigation.

## Conclusion

CANELT, with satisfactory performance in various validity and reliability measures, is recommended to researchers and clinicians as a culturally appropriate and clinically viable tool for assessing the functional communication of Cantonese-speaking PWA.

## Supporting information

**S1 Appendix. Modified CANELT scenarios (translated in English) and reasons for modifications.**
(DOCX)

**S2 Appendix. Scoring criteria of CANELT with examples.**
(DOCX)

**S3 Appendix. Results of pair-wise comparisons of the performance of healthy controls on the two measures of CANELT.**
(DOCX)

## Acknowledgments

I want to express my sincere gratitude to Prof. Sam Po Law for her devotion to initiating and developing the project. Special thanks are also given to Miss Jessica Lo, Elaine Yip, and Emma Lor for their hard work in assisting project development. Finally, I would like to thank the participants enrolled in this study.

## Author Contributions

**Conceptualization:** Winsy Wing Sze Wong.

**Data curation:** Winsy Wing Sze Wong.

**Formal analysis:** Winsy Wing Sze Wong.

**Investigation:** Winsy Wing Sze Wong.

**Methodology:** Winsy Wing Sze Wong.

**Project administration:** Winsy Wing Sze Wong.

**Resources:** Winsy Wing Sze Wong.

**Software:** Winsy Wing Sze Wong.

**Supervision:** Winsy Wing Sze Wong.

**Validation:** Winsy Wing Sze Wong.

**Visualization:** Winsy Wing Sze Wong.

**Writing – original draft:** Winsy Wing Sze Wong.

**Writing – review & editing:** Winsy Wing Sze Wong.

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
