## [Decision Letter · Decision Letter 0]

23 Jan 2024

PONE-D-23-32749Validation of a Cantonese Version of the Amsterdam Nijmegen Everyday Language Test (CANELT): A functional approachPLOS ONE

Dear Dr. Wong,

Thank you for submitting your manuscript to PLOS ONE. After careful consideration, we feel that it has merit but does not fully meet PLOS ONE’s publication criteria as it currently stands. Therefore, we invite you to submit a revised version of the manuscript that addresses the points raised during the review process.

In my capacity as the academic editor, I have thoroughly examined the manuscript and concur with the suggestions provided by the reviewers. It is imperative that you attentively address the recommendations about the methodology and elucidate the rationale behind the indispensability of this study. Both reviewers advocate for revisions to your manuscript to meet the requisite standards for publication. Consequently, I invite you to respond thoughtfully to the reviewers' comments and make the necessary revisions.

We look forward to receiving your revised manuscript.

Kind regards,

Elena Theodorou

Academic Editor

PLOS ONE

Journal Requirements:

Reviewers' comments:

Reviewer's Responses to Questions

**Comments to the Author**

1. Is the manuscript technically sound, and do the data support the conclusions?

Reviewer #1: Partly

Reviewer #2: Yes

2. Has the statistical analysis been performed appropriately and rigorously? 

Reviewer #1: Yes

Reviewer #2: Yes

3. Have the authors made all data underlying the findings in their manuscript fully available?

Reviewer #1: Yes

Reviewer #2: Yes

4. Is the manuscript presented in an intelligible fashion and written in standard English?

Reviewer #1: Yes

Reviewer #2: Yes

5. Review Comments to the Author

Reviewer #1: The study validated the Cantonese version of the Amsterdam-Nijmegen Everyday Language Test (CANELT) for individuals with aphasia. Using quantitative scoring, it assessed communication abilities in both aphasic and healthy individuals. Responses from 56 people with aphasia and 100 healthy Cantonese speakers were orthographically transcribed and analyzed for completeness and information. The author provides a detailed description of the task, its real-life application, and compares it with other available assessments like ASHA FACS.

The differences between ANELT and CANELT lack clarity, and the rationale for using CANELT over ANELT needs further explanation.

The present study demonstrates the adaptation of CANELT for Cantonese speakers, combining data from current and previous studies. However, it's unclear if the same protocol was consistently applied across these studies for test administration and results' analysis. Clarity is needed regarding the re-analysis of orthographic transcriptions from previous studies and any considerations for inter-rater consistency in transcription in both present and previous studies.

Further information is necessary regarding participant recruitment and diagnosis. Were assessments conducted by registered speech pathologists, psychologists, or neurologists? How were neurodegenerative conditions ruled out?

Concerning Table 1, while the left column presents data from healthy adults and the right from people with aphasia, the final row seems to reference fluent/non-fluent aphasia in PWA, with specific aphasia types listed on the right. To avoid confusion, the table format could be split into two or restructured for clarity, as the current presentation may be misleading.

The results' presentation justifies the statistical analysis conducted.

Detailed procedures for translation and cultural adaptation from Dutch to Cantonese are provided.

Overall, the study significantly contributes to cross-linguistic research and the expansion of culturally adapted, language-specific assessment tools

Reviewer #2: In paper Validation of a Cantonese Version of the Amsterdam Nijmegen Everyday Language Test (CANELT): A functional approach autor test the psychometric properties of the ANELT in Cantonese. The paper is well written and the way the data is analysed is very interesting, i.e. it differs significantly from similar studies. Namely, the author provides an overview of how the Cantonese authors develop the scoring system for the CANELT. Analyses based on main concepts provide direct information about the informativeness of discourse production at the macrostructure level as well as opening and new information at the propositional level. In this way, the authors have developed a comprehensive approach to analysing discourse processing and production, which is a prerequisite for effective functional communication. I can recommend the publication of the paper. I have only a few comments that should be taken into account:

Line 61: Can you conclude the sentence with an example of some of the tools; among them, please mention the Scenario Test, as you mention it later in the manuscript. The same sentence – several assessment tools for measuring discourse production…. – actually, there should be - several assessment instruments to measure functional communication based on monologic discourse… because there are many instruments that measure discourse production but not functional communication. For example, in the Boston test or CAT test, there are picture description tasks that measure discourse production but not functional communication.

Line 81 – can you very briefly define Comprehensibility A-Scale and Intelligibilty B-Scale

You use different terms for the control group – neurological healthy individuals, normal controls, healthy normals - ... - please use a single term and use it consistently throughout the manuscript.

In the "Participants" section, provide additional demographic data of PwA – are they all monolingual? What is their level of education? What is their gender?

Line 174 – Indicate the exact number of subjects taken from other authors: ...the rest were based on data collected in other studies: n=? from Law and Lo`s (15), n=? form Yip`s (18) and n=? from Lor`s study (19)

Line 185 – is the Duch-speaking aphasiologist the author of the ANELT? It is important to have the author's permission to translate the test into another language or the publisher's copyright – this is ethically more important than knowing who gave you the test.

Lines 191 and 192 – can you briefly describe five situations that were culturally inappropriate – this may be interesting for readers when confronted with a specific example of cultural bias.

Data analysis section – can you give a very brief and basic definition of each parameter of validity and reliability?

Line 254 – do not use "people with/without aphasia" – when I first read this sentence I thought you were talking about people who have had a stroke but have and do not have aphasia; in fact you are talking about PwA and healthy speakers – this is another example of why it is important that you use terminology consistently.

Lines 266 – 267 – please revise the last part of the sentence - Items with low face validity, i.e., 30% or more respondents rated the item as ‘poor’ in assessing one’s communication ability, were excluded– should be excluded.

Line 349 - which serves as a diagnostic tool – for what? Add: functional communication of Cantonese-speaking PwA

6. PLOS authors have the option to publish the peer review history of their article (what does this mean?). If published, this will include your full peer review and any attached files.

Reviewer #1: No

Reviewer #2: **Yes: **Jelena Kuvač Kraljević

---

## [Author Response · Author response to Decision Letter 0]

5 Mar 2024

Responses to Reviewers

Dear Editor and reviewers, 

Thanks very much for your comments. They are all taken seriously. Please find my responses to each of the comments below. 

Thank you very much and I am looking forward to receiving your reply. 

Best,

Winsy Wong 

Journal Requirements:

Response: The style requirements in the manuscript and file naming have been corrected in the resubmission. 

Response: Thank you for the suggestion. The raw data has been uploaded to OSF (https://osf.io/p3ehr/files/osfstorage/65bf0c659b32ca077097f691). The availability of the raw data file is mentioned in the data availability statement (p. 29; lines 501-502). 

Response: The reference list has been reviewed again. To our understanding no retracted article is cited in the manuscript. 

Reviewers' comments:

Reviewer's Responses to Questions

Comments to the Author

1. Is the manuscript technically sound, and do the data support the conclusions?

Reviewer #1: Partly

Reviewer #2: Yes

2. Has the statistical analysis been performed appropriately and rigorously?

Reviewer #1: Yes

Reviewer #2: Yes

3. Have the authors made all data underlying the findings in their manuscript fully available?

Reviewer #1: Yes

Reviewer #2: Yes

4. Is the manuscript presented in an intelligible fashion and written in standard English?

Reviewer #1: Yes

Reviewer #2: Yes

5. Review Comments to the Author

Reviewer #1: The study validated the Cantonese version of the Amsterdam-Nijmegen Everyday Language Test (CANELT) for individuals with aphasia. Using quantitative scoring, it assessed communication abilities in both aphasic and healthy individuals. Responses from 56 people with aphasia and 100 healthy Cantonese speakers were orthographically transcribed and analyzed for completeness and information. The author provides a detailed description of the task, its real-life application, and compares it with other available assessments like ASHA FACS.

Comment: The differences between ANELT and CANELT lack clarity, and the rationale for using CANELT over ANELT needs further explanation.

Response: The rationale for using CANELT over ANELT is explained on page 7 (lines 116-118). And the differences between ANELT and CANELT are included in p. 10 (lines 164-169). 

Comment: The present study demonstrates the adaptation of CANELT for Cantonese speakers, combining data from current and previous studies. However, it's unclear if the same protocol was consistently applied across these studies for test administration and results analysis. Clarity is needed regarding the re-analysis of orthographic transcriptions from previous studies and any considerations for inter-rater consistency in transcription in both present and previous studies.

Response: The same protocol was consistently applied across studies for test administration (p. 13, lines 225-226) and data analysis (p 15, lines 250-253). To further ensure inter-rater consistency across studies, inter-rater reliability was established based on scores obtained from the two previous studies, i.e., Yip (2019) and Lor (2020). Satisfactory results are obtained and updated in the abstract (p. 3, line 43) and main text (p. 23, line 395). 

Comment: Further information is necessary regarding participant recruitment and diagnosis. Were assessments conducted by registered speech pathologists, psychologists, or neurologists? How were neurodegenerative conditions ruled out?

Response: Assessments were conducted by the author (who is a qualified speech therapist in Hong Kong or by trained speech therapy students under her supervision (p. 11, lines 185-187). Neurodegenerative conditions were based on the medical reports/ reports given by the PWA/caretakers. Such information has been added in the Method Section (p.10-11, lines 181-184)

Comment: Concerning Table 1, while the left column presents data from healthy adults and the right from people with aphasia, the final row seems to reference fluent/non-fluent aphasia in PWA, with specific aphasia types listed on the right. To avoid confusion, the table format could be split into two or restructured for clarity, as the current presentation may be misleading.

Response: Table 1 has been restructured to report the aphasia subtypes of PWA (p.12). 

The results' presentation justifies the statistical analysis conducted.

Detailed procedures for translation and cultural adaptation from Dutch to Cantonese are provided.

Overall, the study significantly contributes to cross-linguistic research and the expansion of culturally adapted, language-specific assessment tools

Reviewer #2: In paper Validation of a Cantonese Version of the Amsterdam Nijmegen Everyday Language Test (CANELT): A functional approach autor test the psychometric properties of the ANELT in Cantonese. The paper is well written and the way the data is analysed is very interesting, i.e. it differs significantly from similar studies. Namely, the author provides an overview of how the Cantonese authors develop the scoring system for the CANELT. Analyses based on main concepts provide direct information about the informativeness of discourse production at the macrostructure level as well as opening and new information at the propositional level. In this way, the authors have developed a comprehensive approach to analysing discourse processing and production, which is a prerequisite for effective functional communication. I can recommend the publication of the paper. I have only a few comments that should be taken into account:

Comment: Line 61: Can you conclude the sentence with an example of some of the tools; among them, please mention the Scenario Test, as you mention it later in the manuscript. The same sentence – several assessment tools for measuring discourse production…. – actually, there should be - several assessment instruments to measure functional communication based on monologic discourse… because there are many instruments that measure discourse production but not functional communication. For example, in the Boston test or CAT test, there are picture description tasks that measure discourse production but not functional communication.

Response: Thank you for the suggestion. Some other functional communication measures, including the Scenario Test, have been included and mentioned (p. 4, lines 62-67). 

Comment: Line 81 – can you very briefly define Comprehensibility A-Scale and Intelligibilty B-Scale

Response: The definitions of both scales are included (p.5; lines 84-86). 

Comment: You use different terms for the control group – neurological healthy individuals, normal controls, healthy normals - ... - please use a single term and use it consistently throughout the manuscript.

Response: The term ‘neurologically healthy controls’ is used to refer to the control participants of the current study throughout the manuscript for consistency. 

Comment: In the "Participants" section, provide additional demographic data of PwA – are they all monolingual? What is their level of education? What is their gender?

Response: The language use and level of education of PWA are included in p. 11; lines 188-190. Gender distribution is included in Table 1 (p.12)

Comment: Line 174 – Indicate the exact number of subjects taken from other authors: ...the rest were based on data collected in other studies: n=? from Law and Lo`s (15), n=? form Yip`s (18) and n=? from Lor`s study (19)

Response: 

The exact number of subjects taken from each study is now included (p. 11, lines 194-196). 

Comment: Line 185 – is the Duch-speaking aphasiologist the author of the ANELT? It is important to have the author's permission to translate the test into another language or the publisher's copyright – this is ethically more important than knowing who gave you the test.

Response: We tried to contact the author of the test but he passed away some time ago. We contacted the publisher and has obtained an agreement ‘Translation from the ANELT/ANTAT, copyright 2008 Hogrefe Uitgevers BC (Amsterdam, the Netherlands) and Cognition Products BV – L. Blomert’ for translating the test and dissemination of research findings (p. 12, lines 206-208) in journal articles. Under the current agreement, we are not allowed to publish, distribute or upload the content of CANELT. Therefore, Supplementary Material 1 (S1) has been modified only to include the five modified scenarios (in their English translated version) and their reasons for the modification. 

Comment: Lines 191 and 192 – can you briefly describe five situations that were culturally inappropriate – this may be interesting for readers when confronted with a specific example of cultural bias.

Response: The five situations where modifications were made are specified in S1 Appendix. 

Comment: Data analysis section – can you give a very brief and basic definition of each parameter of validity and reliability?

Response: A brief definition of each parameter of validity and reliability is given in the respective section (p. 16, lines 270-271, 277-279; p. 16-17, 284-290; p. 17, 297-299,p.18, 309-310, 314-323; p. 19, lines 325-329). 

Comment: Line 254 – do not use "people with/without aphasia" – when I first read this sentence I thought you were talking about people who have had a stroke but have and do not have aphasia; in fact you are talking about PwA and healthy speakers – this is another example of why it is important that you use terminology consistently.

Response: The term ‘neurologically healthy controls’ has been used throughout the manuscript to avoid confusion. 

Comment: Lines 266 – 267 – please revise the last part of the sentence - Items with low face validity, i.e., 30% or more respondents rated the item as ‘poor’ in assessing one’s communication ability, were excluded– should be excluded.

Response: Revision has been made accordingly (p.18, line 307).

Comment: Line 349 - which serves as a diagnostic tool – for what? Add: functional communication of Cantonese-speaking PwA

Response: Revision has been made in the respective content (p. 23, lines 401-402).

---

## [Decision Letter · Decision Letter 1]

17 Apr 2024

PONE-D-23-32749R1Validation of a Cantonese Version of the Amsterdam Nijmegen Everyday Language Test (CANELT): A functional approachPLOS ONE

Dear Dr. Wong,

Thank you for submitting your manuscript to PLOS ONE. After careful consideration, we are pleased to inform you that your work has been found to have merit for publication. However, in order to ensure its readiness for publication, some minor revisions are required, as suggested by one of the reviewers. Specifically, the reviewer has recommended language edits and corrections for typographical errors. These adjustments are standard procedure and will contribute to the overall clarity and readability of your manuscript. We kindly ask that you review the suggested edits and make necessary revisions accordingly. Once these adjustments are completed, we encourage you to resubmit the final version of your paper at your earliest convenience. We appreciate your attention to these revisions and look forward to the continued progress of your manuscript.

We look forward to receiving your revised manuscript.

Kind regards,

Elena Theodorou

Academic Editor

PLOS ONE

Journal Requirements:

Reviewers' comments:

Reviewer's Responses to Questions

**Comments to the Author**

1. If the authors have adequately addressed your comments raised in a previous round of review and you feel that this manuscript is now acceptable for publication, you may indicate that here to bypass the “Comments to the Author” section, enter your conflict of interest statement in the “Confidential to Editor” section, and submit your "Accept" recommendation.

Reviewer #1: All comments have been addressed

Reviewer #2: All comments have been addressed

2. Is the manuscript technically sound, and do the data support the conclusions?

Reviewer #1: Yes

Reviewer #2: Yes

3. Has the statistical analysis been performed appropriately and rigorously? 

Reviewer #1: Yes

Reviewer #2: Yes

4. Have the authors made all data underlying the findings in their manuscript fully available?

Reviewer #1: Yes

Reviewer #2: Yes

5. Is the manuscript presented in an intelligible fashion and written in standard English?

Reviewer #1: Yes

Reviewer #2: Yes

6. Review Comments to the Author

Reviewer #1: Thank you for the opportunity to review the revised version of the manuscript. I acknowledge that all issues raised in the previous review round have been satisfactorily addressed.

Reviewer #2: As I said in my first review, I like the paper because it is not a classical, i.e. rather technical paper that only provides data on the validity and reliability of the test. This paper contains detailed information about the reasons (motivations) for the development of the CANELT, its advantages and disadvantages and, most importantly, about all the innovations that various authors have introduced over the last five years of its development to improve it in part of the test items and in the scoring section.

I have only a few small corrections:

Line 66: I would not say: among these measures …rather say Among these assessment tools…

In general, I would not speak of normative performance, since performance is always variable and therefore needs to be normed – I would replace this syntagm normative performance accordantly in the paper with normative data (e.g. lines 192, 261, 267, etc).

The sentence in lines 68-70 has the same content as the sentence in lines 80-82. Delete the sentence in lines 80-82.

Line 86: explanation of B-Scale is somewhat unclear – I recommend that you add…examines the perception of utterances in such a way that words are recognizable regardless of their meaning.

Line 158 – add reference number [24] after Lor

Line 186 – please use the terminology speech-language therapist and use this term accordingly (e.g. in line 223).

Line 223 – add trained – who was either a trained speech-language therapy student…

Line 187 – delete therapy – The trained students had ….

Line 272 – add neurologically – among neurologically healthy controls…

Lines 341 -351 – the results show a decrease in performance on the variables Opening and New Information in both groups – it would be interesting to hear the author’s explanation for these results, especially for neurologically healthy controls, in the discussion section. (*This is just an idea for a new study - it would be intersting to analyse the data from neurologically healthy controls in two age groups only – for example, younger adults (30 to 59 years) and older (60+ years), as the data presented in this paper shows a trend of performance change around age 60. This aligns with Lor`s idea in lines 158-161, who attempted to develop age-based norms – perhaps with such a wide range of age groups, age differences can be captured).

7. PLOS authors have the option to publish the peer review history of their article (what does this mean?). If published, this will include your full peer review and any attached files.

Reviewer #1: No

Reviewer #2: **Yes: **Jelena Kuvač Kraljević

---

## [Author Response · Author response to Decision Letter 1]

22 Apr 2024

Journal Requirements:

Response: The reference list has been reviewed. 

Reviewers' comments:

Reviewer's Responses to Questions

Comments to the Author

1. If the authors have adequately addressed your comments raised in a previous round of review and you feel that this manuscript is now acceptable for publication, you may indicate that here to bypass the “Comments to the Author” section, enter your conflict of interest statement in the “Confidential to Editor” section, and submit your "Accept" recommendation.

Reviewer #1: All comments have been addressed

Reviewer #2: All comments have been addressed

2. Is the manuscript technically sound, and do the data support the conclusions?

Reviewer #1: Yes

Reviewer #2: Yes

3. Has the statistical analysis been performed appropriately and rigorously?

Reviewer #1: Yes

Reviewer #2: Yes

4. Have the authors made all data underlying the findings in their manuscript fully available?

Reviewer #1: Yes

Reviewer #2: Yes

5. Is the manuscript presented in an intelligible fashion and written in standard English?

Reviewer #1: Yes

Reviewer #2: Yes

6. Review Comments to the Author

Reviewer #1: Thank you for the opportunity to review the revised version of the manuscript. I acknowledge that all issues raised in the previous review round have been satisfactorily addressed.

Reviewer #2: As I said in my first review, I like the paper because it is not a classical, i.e. rather technical paper that only provides data on the validity and reliability of the test. This paper contains detailed information about the reasons (motivations) for the development of the CANELT, its advantages and disadvantages and, most importantly, about all the innovations that various authors have introduced over the last five years of its development to improve it in part of the test items and in the scoring section.

I have only a few small corrections:

Line 66: I would not say: among these measures …rather say Among these assessment tools…

Response: The suggested amendment has been made (p. 4, line 66). 

In general, I would not speak of normative performance, since performance is always variable and therefore needs to be normed – I would replace this syntagm normative performance accordantly in the paper with normative data (e.g. lines 192, 261, 267, etc).

Response: The term ‘normative data’ has been used to replace ‘normative performance’ (lines 69, 192, 256-257, 261, 267). 

The sentence in lines 68-70 has the same content as the sentence in lines 80-82. Delete the sentence in lines 80-82.

Response: The sentence in lines 80-82 has been deleted (p. 5, lines 80-82). 

Line 86: explanation of B-Scale is somewhat unclear – I recommend that you add…examines the perception of utterances in such a way that words are recognizable regardless of their meaning.

Response: The suggested change has been made (p. 5, lines 86-87). 

Line 158 – add reference number [24] after Lor

Response: Reference number [24] has been added after Lor (p. 9, line 158). 

Line 186 – please use the terminology speech-language therapist and use this term accordingly (e.g., in line 223).

Response: The terminology ‘speech-language therapist’ has been used consistently in the manuscript (p. 11, line 186; p. 13, line 223-224). 

Line 223 – add trained – who was either a trained speech-language therapy student…

Response: The word ‘trained’ has been added (p. 13, line 223). 

Line 187 – delete therapy – The trained students had ….

Response: The word ‘therapy’ has been deleted (p. 11, line 187). 

Line 272 – add neurologically – among neurologically healthy controls…

Response: The word ‘neurologically’ has been added (p. 16, line 272). 

Lines 341 -351 – the results show a decrease in performance on the variables Opening and New Information in both groups – it would be interesting to hear the author’s explanation for these results, especially for neurologically healthy controls, in the discussion section. (*This is just an idea for a new study - it would be intersting to analyse the data from neurologically healthy controls in two age groups only – for example, younger adults (30 to 59 years) and older (60+ years), as the data presented in this paper shows a trend of performance change around age 60. This aligns with Lor`s idea in lines 158-161, who attempted to develop age-based norms – perhaps with such a wide range of age groups, age differences can be captured).

Response: The decrease in performance in CANELT and its possible relation with cognitive decline associated with aging has been included in the Discussion (p. 27-28, lines 472-496). However, the aging effect observed in the current study should be treated with caution, given its limitations (p. 27, lines 481-485). It is suggested that such a topic should deserve further investigation (p. 28, lines 492-496).

---

## [Editor Report · Decision Letter 2]

1 May 2024

Validation of a Cantonese Version of the Amsterdam Nijmegen Everyday Language Test (CANELT): A functional approach

PONE-D-23-32749R2

Dear Dr. Wong 

We’re pleased to inform you that your manuscript has been judged scientifically suitable for publication and will be formally accepted for publication once it meets all outstanding technical requirements.

Kind regards,

Elena Theodorou

Academic Editor

PLOS ONE